# Transforming a fragile protein helix into an ultrastable scaffold via a hierarchical AI and chemistry framework

**Jun Qiu, Guojin Tang, Tianfu Feng, Bin Zheng, Yuanhao Liu, Peng Zheng***

State Key Laboratory of Coordination Chemistry, Nanjing Drum Tower Hospital, Affiliated Hospital of Medical School, School of Chemistry and Chemical Engineering, Chemistry and Biomedicine Innovation Center (ChemBIC), Frontier Interdisciplinary Science Research Center, Nanjing University, Nanjing, China

## eLife Assessment

This **important** work describes systematic computational and experimental approaches to turn a moderately stable α-helical bundle into a very stable fold. The authors advance our understanding of α-helix stabilization providing a convenient framework that has general implications for the protein design field. The main claims have **convincing** support through a sound methodology, with strong specific conclusions for designing mechanically, thermally, and chemically stable α-helical bundles.

***For correspondence:**
pengz@nju.edu.cn

**Competing interest:** The authors declare that no competing interests exist.

**Abstract** The rational design of proteins that maintain structural integrity under concurrent thermal, mechanical, and chemical stress remains a challenge in molecular engineering. We present a hierarchical framework that transforms an α-helical domain into an ultrastable scaffold by integrating AI-guided design with foundational chemical principles. This approach progresses from global architectural reinforcement, using multiple AI tools to create a stabilized four-helix bundle, to local chemical tuning, where AlphaFold3 guides the installation of salt bridges and metal-coordination motifs. A computational pipeline using physics-based screening such as molecular dynamics simulations efficiently distilled millions of designs into a minimal candidate set. The resulting α-helical proteins exhibit unprecedented multi-axis stability, with mechanical unfolding forces exceeding 200 pN, thermal resilience>100°C, and high resistance to chemical denaturants. By systematically dissecting the contributions of hydrophobic packing, electrostatics, and metal coordination, we establish a general blueprint for imparting extreme robustness. This work bridges AI-driven structural generation with chemical precision, advancing the creation of durable proteins for mechanistic studies and synthetic biology.

## Introduction

Proteins that maintain their structural integrity when heated, stretched, or exposed to harsh solvents are foundational to both life (*Manteca et al., 2017*; *Neupane et al., 2024*; *Rief et al., 1997*) and serve as invaluable building blocks for advanced biomaterials (*Fu et al., 2023*; *Miserez et al., 2023*; *Sivasankaran et al., 2024*), molecular devices (*Devaux et al., 2021*; *Yang et al., 2024*), and operations in extreme environments (*Bustamante et al., 2004*; *Pepelnjak et al., 2024*). Among natural scaffolds, helical architectures are particularly appealing targets due to their ubiquity and mechanical functionality: coiled-coils contribute to muscle and other cytoskeletal assemblies (*Baumann et al., 2017*; *Schwaiger et al., 2002*), talin rod domains act as mechanosensors that unfold under load (*del Rio et al., 2009*; *Le et al., 2019*; *Tapia-Rojo et al., 2023*), and arrays of α-helical bundles underlie

the erythrocyte membrane skeleton (*Rief et al., 1999*; *Takahashi et al., 2018*; *Wensley et al., 2010*). These precedents motivate the goal of transforming common α-helical folds into programmable components with superior, multi-axis stability for diverse applications.

However, the rational design of such multi-faceted stability into isolated helical domains remains challenging (*Baker et al., 2017*). The inherent mechanical fragility of the α-helix is a fundamental limitation; under tensile load, its hydrogen bonds rupture sequentially, leading to low unfolding forces of just tens of piconewtons (pN) (*Carrion-Vazquez et al., 2000*; *Hoffmann et al., 2013*; *Law et al., 2003*; *LeBlanc et al., 2021*; *Müller et al., 2021*; *Neuman and Nagy, 2008*; *Randles et al., 2007*), well below the hundreds of pN reached by mechanically robust β-sheet topologies pulled in shear (*Brockwell et al., 2003*). This inherent fragility persists even in the presence of stable hydrophobic cores (*Collet et al., 2005*; *Craig et al., 2004*; *Lee et al., 2006*; *Lim et al., 2008*; *Milles et al., 2017*; *Minin et al., 2017*; *Ng et al., 2007*).

While decades of research have identified key chemical determinants of protein stability like hydrophobic packing, electrostatics, and metal coordination, with both natural and engineered examples showing these factors can enhance resilience (*Baker et al., 2015*; *Goldenzweig and Fleishman, 2018*; *Kellis et al., 1988*; *Marqusee and Baldwin, 1987*; *Nandi and Ainavarapu, 2022*; *Wang et al., 2021*; *Zhang et al., 2025*), progress in increasing mechanical strength of isolate helical proteins has been a meaningful yet incremental process (*López-García et al., 2021*; *Tunn et al., 2018*). For example, only a few natural coiled-coils (e.g., in fibrinogen) approach 100 pN, while canonical α-helical bundles such as spectrin and ankyrin repeats typically unfold near 50 pN under comparable loading. The key challenge we address is thus to move beyond adding individual stabilizing features, and instead to precisely combine these different chemical interactions within a single architecture so they work together to resist multiple forms of stress.

The recent rise of generative AI for protein design has dramatically expanded the accessible fold space (*Dauparas et al., 2022*; *Jumper et al., 2021*; *Lin et al., 2023*; *Sakuma et al., 2024*; *Watson et al., 2023*), offering a path to overcome the architectural limitations of natural scaffolds. However, a critical gap persists. AI models excel at global structure generation but often lack the understanding of local chemical interaction networks required for extreme, multi-axis stability. These models generate vast candidate libraries, creating a screening bottleneck that impedes the experimental validation essential for iterative, chemically rational design. A method that seamlessly integrates generative design with highly efficient screening is needed to bridge this gap.

Here, we treat multi-axis protein stabilization as a problem of hierarchical chemical engineering. We introduce an AI-guided blueprint that progresses systematically from global architectural reinforcement to atomistically precise chemical installation. First, AI-enabled backbone construction generates stable scaffolds with optimized hydrophobic cores, establishing a foundational stability layer. We then apply precision functionalization, installing a minimal set of inter-helical salt bridges and bi-histidine metal-coordination motifs through rational, site-specific edits. This stepwise approach allows us to quantitatively dissect the mechanical contribution of each engineered chemical interaction (*Sakuma et al., 2024*).

To make this hierarchical chemical design tractable, we implement a computational pipeline that leverages developability filters (*Kyte and Doolittle, 1982*), foldability assessment (*Jumper et al., 2021*; *Lin et al., 2023*), and critically molecular dynamics (MD) simulations (*Karplus and McCammon, 2002*; *Milles et al., 2018*; *Rennekamp et al., 2024*) to rank candidates based on proxies for mechanical and thermal resilience. This pipeline compresses ~$10^6$ in silico designs to a manageable number for experimental validation. The result is a family of ultrastable α-helical proteins where we can quantitatively attribute stability gains to specific chemical engineering steps: hydrophobic core optimization, electrostatic interaction, and metal-coordination clamping. This work provides a general, rational blueprint for the precision engineering of protein energy landscapes, transforming a mechanically weak architecture into a durable platform.

## Results

### Hierarchical design of ultrastable proteins by integrating AI and chemical principles

To establish a framework for programming multi-axis stability, we developed a hierarchical design strategy that integrates architectural and chemical stabilization principles. We selected the human spectrin repeat R15 (PDB code: 3F57) (*Ipsaro et al., 2009*), a three-helix bundle that unfolds at ~50 pN and melts near 50°C, as our design template (*Figure 1a, b*). These moderate baseline properties provide an ideal starting point for quantitatively evaluating the contribution of each engineered stabilization layer. Our approach comprises two distinct stages (*Figure 1c*): Stage I establishes a stabilized architectural framework through AI-guided backbone construction and computational screening, while Stage II applies precision functionalization with site-specific salt bridges and metal-coordination motifs to reinforce key structural interfaces.

In Stage I, we expanded the native architecture by appending a fourth helix to the spectrin template using RFdiffusion (*Figure 1d*; *Watson et al., 2023*). We generated 100 backbone variants (50 each for N- and C-terminal extensions) and selected five optimal four-helix scaffolds based on bundle geometry, helical registry, and the potential for hydrophobic core densification without steric clash (*Lombardi et al., 2019*). For each selected scaffold, we used ProteinMPNN to generate 100,000 sequences optimized for the remodeled core, producing a total library of $5 \times 10^5$ candidate sequences (*Liu et al., 2025*).

To identify the best designs fulfilling our stability criteria, we implemented a multi-stage computational funnel that progresses from coarse filtering to high-fidelity functional prediction (*Figure 1e*). The library was first subjected to a developability screen using physicochemical criteria (GRAVY <−0.3), retaining about 300,000 sequences with high predicted solubility and yield. Foldability was then assessed through a two-tiered model: ESMFold provided an initial screening to 357 designs, which were subsequently refined with AlphaFold2 under strict confidence thresholds (pLDDT >90, root mean square deviation [RMSD] <2.0 Å). This process yielded 211 high-confidence designs for further analysis.

Structural characterization revealed that the designed proteins exhibited enhanced packing, with a lower relative solvent-accessible surface area (*Tien et al., 2013*) (rSASA ≈ 0.35) compared to the natural spectrin repeat (rSASA ≈ 0.40). This was further supported by a higher proportion of residues with rSASA <0.2 (designs: >30% vs. natural spectrin: 23%, see Raw Data 1), indicating successful hydrophobic core optimization (*Tien et al., 2013*).

We then implemented MD simulations to prioritize candidates based on functional stability metrics. Steered MD (SMD) simulations served as a mechanical proxy, stretching each domain under constant-velocity pulling to obtain relative stability rankings (*Figure 1b*; *Ni et al., 2024*). Complementary annealing MD (AMD) simulations assessed thermal resilience through temperature ramping (310–473 K) while monitoring structural integrity. These physics-based simulations provided the discriminatory power necessary to select candidates exhibiting target multi-axis stability. By integrating SMD and AMD results, we selected a final experimental shortlist that balances predicted mechanical and thermal resilience.

From this refined set, three variants, originally designated SpecAI87788, SpecAI16941, and SpecAI95889 (*Figure 1—figure supplement 1*) based on their AI-generated identifiers, were selected for experimental validation based on their superior combined SMD and AMD performance (*Figure 1—figure supplement 2* and *Figure 1—figure supplement 3*, *Figure 1—videos 1–10*). For clarity in subsequent analyses, these were renamed SpecAI88, SpecAI41, and SpecAI89, reflecting the last two digits of their original IDs.

The funnel compressed ~$10^6$ AI-generated sequences to three best candidates, preserving both scaffold and sequence diversity while prioritizing top performers. This selection process demonstrates how computational methods can efficiently navigate vast design spaces to identify optimal candidates for experimental characterization.

### Experimental verification of multi-axis stability in designed proteins

From the computational shortlist, we selected three representative designs, SpecAI88, SpecAI41, and SpecAI89, for experimental validation. All three expressed soluble in *E. coli* with high yields (~500 mg/l in TB, *Table 1*). SDS–PAGE showed the expected band near 19 kDa (*Figure 2—figure supplement*

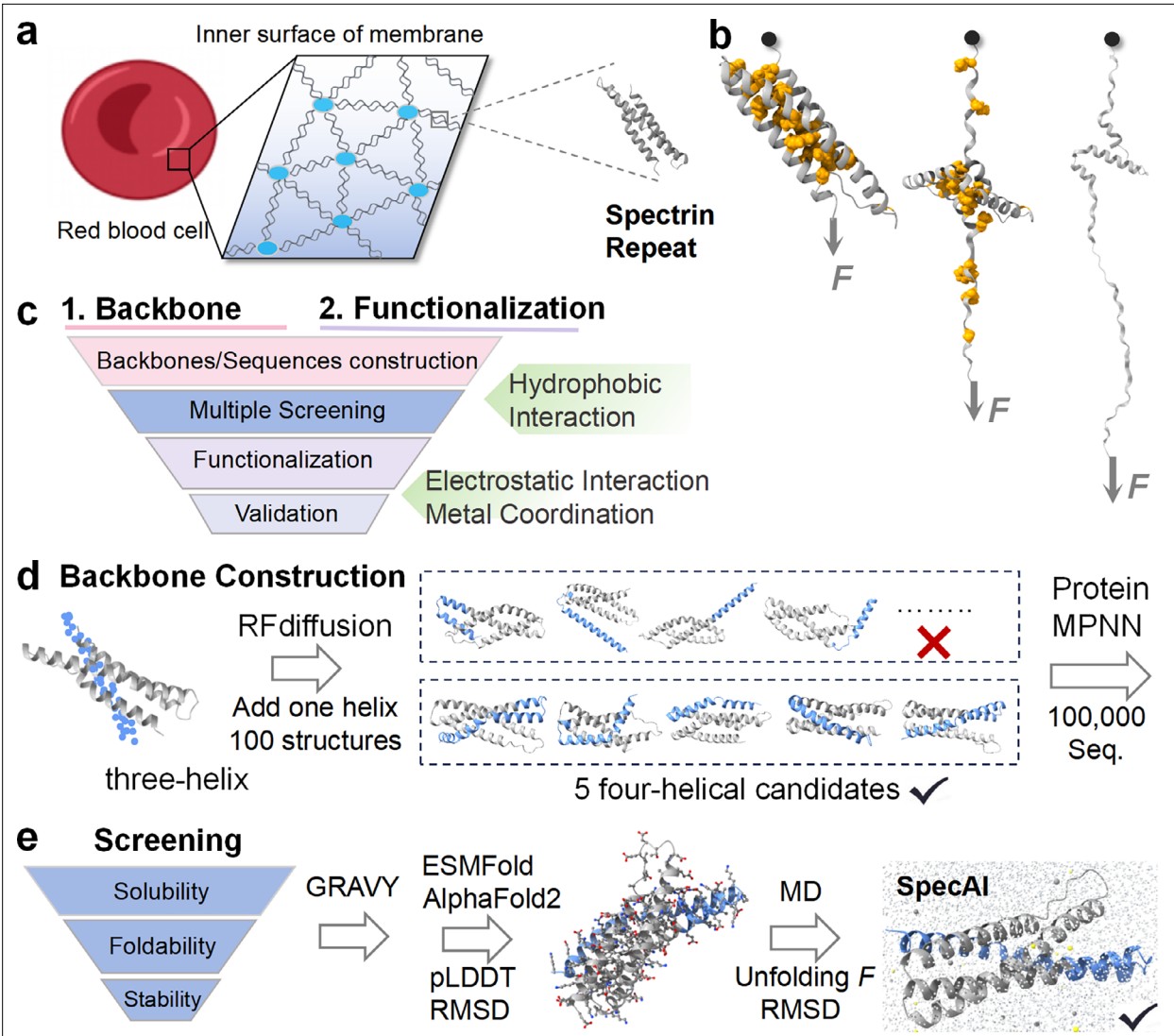

**Figure 1.** A hierarchical chemical blueprint for multi-axis stabilization of α-helical protein. (**a**) The erythrocyte membrane skeleton, which relies on spectrin repeats to withstand shear stress. Inset: Structure of a single, naturally fragile spectrin repeat (PDB 3F57), with hydrophobic core residues shown. (**b**) The mechanical unfolding pathway of a spectrin repeat under tensile force, probed computationally by steered molecular dynamics (SMD) to generate force–extension curves. (**c**) Overview of the two-stage design strategy. Stage I (architectural stabilization): AI-guided backbone construction and computational screening generate four-helix designs with optimized hydrophobic cores. Stage II (precision functionalization): Rational installation of inter-helical salt bridges and metal-coordination motifs reinforces specific mechanical interfaces. (**d**) Stage I, backbone construction. RFdiffusion appends a fourth helix to the native three-helix template, generating 100 initial backbones. Five optimal four-helix scaffolds are selected, and ProteinMPNN is used to generate 100,000 sequences per scaffold. (**e**) Stage I, computational screening. A multi-step funnel prioritizes candidates through successive filters: developability (GRAVY score ≤−0.3), foldability (ESMFold triage followed by AlphaFold2 refinement with root mean square deviation [RMSD] ≤2.0 Å and pLDDT ≥90), and stability assessed via molecular dynamics. The process efficiently narrows ~$10^6$ initial designs to an experimentally tractable shortlist.

The online version of this article includes the following video and figure supplement(s) for figure 1:

**Figure supplement 1.** 2D structure and sequence of the designed protein.

**Figure supplement 2.** The steered molecular dynamics (SMD) simulation results of natural spectrin, SpecAI88, SpecAI41, and SpecAI89.

**Figure supplement 3.** Thermal stability of natural spectrin and SpecAI designs via annealing molecular dynamics (MD).

**Figure 1—video 1.** Steered molecular dynamics (SMD) video of Spectrin unfolding under mechanical pulling.
https://elifesciences.org/articles/109753/figures#fig1video1

**Figure 1—video 2.** Steered molecular dynamics (SMD) video of SpecAI88 unfolding under mechanical pulling.
https://elifesciences.org/articles/109753/figures#fig1video2

*Figure 1 continued on next page*

*Figure 1 continued*

**Figure 1—video 3.** Steered molecular dynamics (SMD) video of SpecAI41 unfolding under mechanical pulling.
https://elifesciences.org/articles/109753/figures#fig1video3

**Figure 1—video 4.** Steered molecular dynamics (SMD) video of SpecAI89 unfolding under mechanical pulling.
https://elifesciences.org/articles/109753/figures#fig1video4

**Figure 1—video 5.** Molecular dynamics (MD) video of Spectrin and SpecAI series unfolding under mechanical pulling.
https://elifesciences.org/articles/109753/figures#fig1video5

**Figure 1—video 6.** Annealing molecular dynamics (AMD) video of Spectrin upon thermal denaturation.
https://elifesciences.org/articles/109753/figures#fig1video6

**Figure 1—video 7.** Annealing molecular dynamics (AMD) video of SpecAI88 upon thermal denaturation.
https://elifesciences.org/articles/109753/figures#fig1video7

**Figure 1—video 8.** Annealing molecular dynamics (AMD) video of SpecAI41 upon thermal denaturation.
https://elifesciences.org/articles/109753/figures#fig1video8

**Figure 1—video 9.** Annealing molecular dynamics (AMD) video of SpecAI89 upon thermal denaturation.
https://elifesciences.org/articles/109753/figures#fig1video9

**Figure 1—video 10.** Annealing molecular dynamics (AMD) video of Spectrin and SpecAI series upon thermal denaturation.
https://elifesciences.org/articles/109753/figures#fig1video10

*1*), and mass spectroscopy confirmed molecular weights of 19,760, 19,108, and 19,285 Da, respectively (*Figure 2a*). Far-UV circular dichroism (CD) spectroscopy was performed, revealing that all three proteins adopted predominantly α-helical structures, as indicated by the characteristic double minima at 208 and 222 nm (*Figure 2b*; *Greenfield, 2006*).

To evaluate thermal stability, we performed temperature-dependent CD measurements from 20 to 100°C. All three designed proteins maintained substantial helical structure at 100°C, as evidenced by the retention of characteristic ellipticity at 195 nm, demonstrating exceptional thermodynamic stability (*Figure 2b*, inset). We further challenged the proteins by performing thermal denaturation in the presence of the chemical denaturant guanidine hydrochloride (GdnHCl) (*Liu and Thirumalai, 2025*). Notably, all designs preserved their secondary structure at denaturant concentrations up to 3 M and maintained significant helical content under elevated temperature conditions (~50°C, 222 nm), demonstrating remarkable resistance to combined chaotropic stress (*Figure 2c*). These results confirm not only the proper folding of these monomeric α-helical scaffolds but also validate that our Stage I designs, featuring optimized hydrophobic cores, achieve exceptional stability against both thermal and chemical denaturation.

Next, we directly quantified their mechanical stability using atomic force microscopy-based single-molecule force spectroscopy (AFM–SMFS). Each designed SpecAI domain was fused in series with three GB1 fingerprint domains and pulled via the high-strength Coh–Doc handle (rupture force ~400 pN), enabling unambiguous single-molecule identification (*Figure 3a*; *Deng et al., 2019*; *Shi et al., 2022*; *Stahl et al., 2012*). All experiments were conducted in a standard buffer at a pulling speed of 1000 nm/s unless otherwise specified. Force–extension curves were fitted with the worm-like chain (WLC) model to determine the contour length increment (ΔLc) for each unfolding event (*Marko and Siggia, 1995*).

As expected, the natural spectrin control (102 amino acids in length) exhibited a ΔLc of 32 nm, in addition to the unfolding steps of the three GB1 domains (18 nm each) (*Cao et al., 2006*). In contrast, the designed SpecAI proteins showed a significantly larger ΔLc of ≈52 nm (*Figure 3b*,

**Table 1.** Stability of AI-designed spectrin with four α-helices.

| Protein | SMD (pN) 1 nm/ns | Tm (°C) | Yield (mg/l) | AFM (pN) | ΔLc (nm) | ΔF (pN) |
|---|---|---|---|---|---|---|
| Spectrin | 599 ± 62 | ~50 | ~50 | 56 ± 3 | 32 ± 1 | N/A |
| SpecAI88 | 735 ± 80 | >100 | 580 ± 36 | 116 ± 2 | 52 ± 2 | 60 |
| SpecAI41 | 689 ± 84 | >100 | 625 ± 28 | 156 ± 4 | 53 ± 1 | 100 |
| SpecAI89 | 670 ± 84 | >100 | 450 ± 16 | 121 ± 4 | 52 ± 2 | 61 |

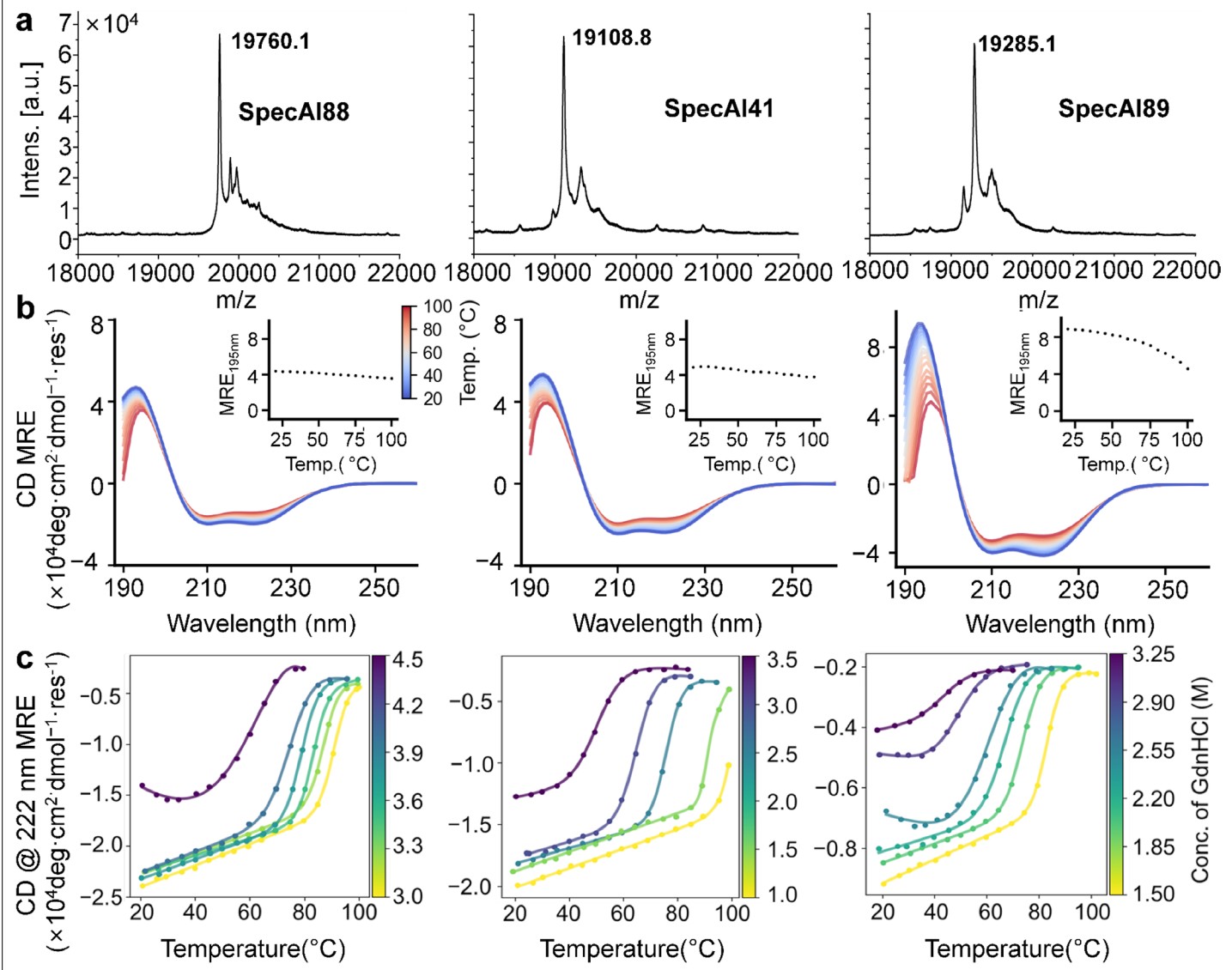

**Figure 2.** Thermal and chemical robustness of AI-designed spectrin variants. (**a**) MALDI-TOF mass spectrometry confirms molecular weights for SpecAI88, SpecAI41, and SpecAI89. (**b**) Far-UV circular dichroism (CD; 190–260 nm) shows α-helical signatures with minima at 208 and 222 nm. Temperature-dependent CD (20–100°C, blue to red) indicates substantial retention of ellipticity at 195 nm, with melting temperatures exceeding 100°C. (**c**) CD at 222 nm recorded in GdnHCl demonstrates persistence of α-helical signal at high denaturant concentrations (~3 M), indicating high chemical resistance.

The online version of this article includes the following source data and figure supplement(s) for figure 2:

**Figure supplement 1.** SDS–PAGE results of SpecAI variants, show their successful expression with expected MW of ~19 *k*Da.

**Figure supplement 1—source data 1.** Original SDS-PAGE gel.

**Figure supplement 1—source data 2.** Original SDS-PAGE gel, indicating the relevant bands.

*Figure 1—figure supplement 1*), consistent with the full unraveling of their ~150-residue domain. The measured values: 52 ± 2 nm (SpecAI88), 53 ± 1 nm (SpecAI41), and 52 ± 2 nm (SpecAI89) (mean ± standard error of the mean [SEM], from Gaussian fit), closely matched theoretical predictions, thereby confirming the correct assignment of the designed domain's unfolding (*Figure 3c*).

Most importantly, all three SpecAI variants exhibited markedly elevated unfolding forces compared to natural spectrin (56 ± 3 pN; *n* = 212). The unfolding forces reached 116 ± 2 pN (SpecAI88; *n* = 224), 156 ± 4 pN (SpecAI41; *n* = 218), and 121 ± 4 pN (SpecAI89; *n* = 174) (*Figure 3c*, *Table 1*). The combination of accurate ΔLc, the presence of three GB1 fingerprint domains, and consistent force statistics

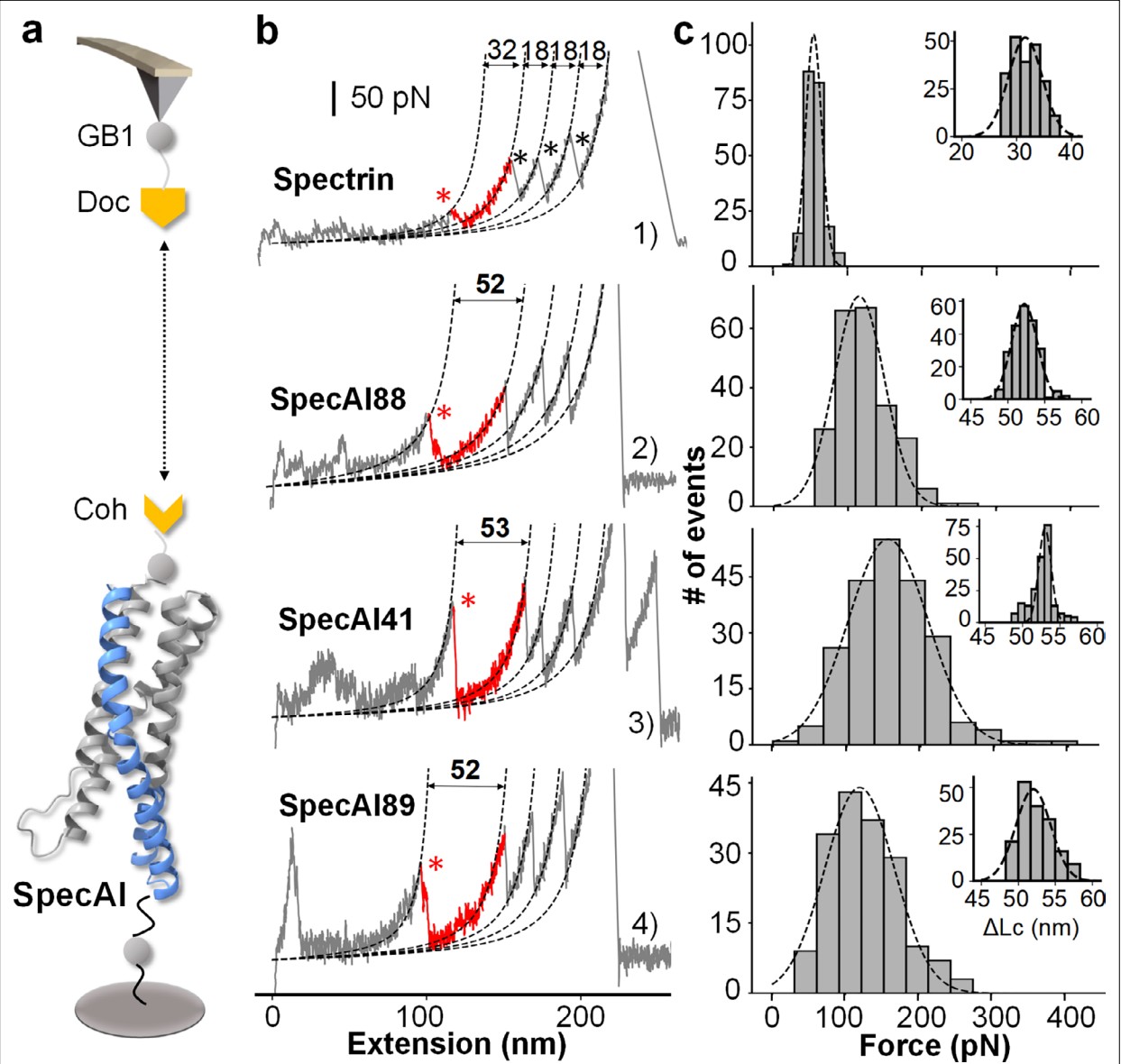

**Figure 3.** Stage I four-helix bundle designs exhibit enhanced mechanical stability by single-molecule force spectroscopy. (**a**) Schematic of the AFM–SMFS experimental setup. A dockerin (Doc)-functionalized AFM tip engages a cohesin (Coh)-tagged SpecAI construct immobilized on the surface, enabling single-molecule pulling. The construct includes three GB1 domains as fingerprint markers. (**b**) Representative force–extension curves. Curve 1: Unfolding of the natural spectrin repeat (ΔLc ≈ 32 nm, red), followed by three GB1 (ΔLc ≈ 18 nm per domain, black). Curves 2–4: Unfolding of SpecAI variants (ΔLc ≈ 53 nm). Sometimes, the Doc unfolds, showing additional peak (Curve 3). Dashed lines are worm-like chain model fit. (**c**) Unfolding force histograms of SpecAI (bin size = 30 pN) with Gaussian fits demonstrate a significant increase in mechanical stability compared to the native spectrin repeat (56 ± 3 pN, $n$ = 212, bin size = 13.75 pN): SpecAI88, 116 ± 2 pN ($n$ = 224); SpecAI41, 156 ± 4 pN ($n$ = 218); and SpecAI89, 121 ± 4 pN ($n$ = 174). The corresponding ΔLc distributions are centered near 52 nm: SpecAI88, 52 ± 2 pN ($n$ = 224); SpecAI41, 53 ± 1 pN ($n$ = 218); and SpecAI89, 52 ± 2 pN ($n$ = 174), consistent with the full unfolding of the designed domain.

confirms single-molecule specificity and establishes a new mechanical baseline for α-helical proteins created through Stage I backbone design.

Values for SMD are reported as mean ± SD, while all other values are reported as Gaussian fit mean ± SEM. ΔF means the unfolding force increase over spectrin.

## Precision scaffold functionalization via hierarchical chemical design

Building upon the stabilized architectural frameworks from Stage I, we implemented a precision functionalization strategy to further enhance stability through the rational design of specific chemical

interactions at inter-helical interfaces. This stage involved two complementary approaches: engineering of electrostatic salt bridges and incorporation of metal-coordination motifs (*Figure 4a*).

Our design strategy followed a rigorous hierarchical selection process. First, we identified potential mutation sites by excluding residues critical for hydrophobic core integrity, focusing instead on positions at inter-helical interfaces that could accommodate new interactions without perturbing the existing structural stability. Second, we employed AlphaFold3 (AF3) structure prediction to evaluate potential mutations, requiring that designs not only maintain high overall structural confidence pLDDT >80 for all mutated residues, with many exceeding 90 but also satisfy precise geometric criteria for the intended molecular interactions, specifically in terms of inter-atomic distances.

We designed and incorporated single inter-helical salt bridges into each optimized parent scaffold: SpecAI41-9K152D (*Figure 4b*, *Figure 1—figure supplement 1*), SpecAI88-49E102K, and SpecAI89-25E48K (*Figure 4—figure supplement 3*). AF3 models confirmed that the engineered residues were positioned at interface-facing turns with favorable i ↔ i' or i ± 1 helical register arrangements without introducing steric clashes. Crucially, all designed salt bridges showed charged atom distances <0.4 nm in the predicted structures, and the contacts remained stable following backbone relaxation simulations. This consistency indicates genuine local interface reinforcement rather than global structural reorganization. All mutated backbone atoms maintained high pLDDT scores (>80), confirming the preservation of structural integrity. For example, the pLDDT scores for the $C\alpha$ atoms were 90.1 and 80.7 for SpecAI41-9K152D.

AFM–SMFS measurements revealed ~30 pN increases relative to Stage I parents with $\Delta Lc$ unchanged (~53 nm), as expected. Unfolding forces were 180 ± 4 pN ($n$ = 197) for SpecAI41-9K152D (*Figure 4c, d*), 141 ± 3 pN ($n$ = 202) for SpecAI88-49E102K, and 153 ± 4 pN ($n$ = 264) for SpecAI89-25E48K (*Figure 4—figure supplement 3*, *Table 2*). The constant $\Delta Lc$ across parents and salt-bridge variants indicates that electrostatic installation does not change topology or domain length, but raises the mechanical barrier along the established pathway.

To achieve additional mechanical reinforcement, we designed bi-histidine motifs capable of forming intra-domain Ni(II)-coordination sites[63]. These motifs were strategically positioned on adjacent helices to create bidentate metal-binding pockets without perturbing the hydrophobic core. Our design criteria targeted Ni-N(His) distances of ~2.0 Å, $C\alpha$–$C\alpha$ spacing compatible with inter-helical bridging rotamers, and proper side-chain orientation (Nδ1/Nε2) for optimal imidazole coordination geometry. This approach yielded three metal-coordination variants: SpecAI41-9K152D-6H153H, SpecAI88-49E102K-6H149H, and SpecAI89-25E48K-24H135H (*Figure 4*, *Figure 1—figure supplement 1*, and *Figure 4—figure supplement 3*), whose structures were validated using AF3 again.

In the presence of 200 µM $Ni^{2+}$, AFM–SMFS revealed a further significant increase in mechanical stability over the salt-bridge parents, with unfolding forces reaching 208 ± 3 pN ($n$ = 222) for SpecAI41-9K152D-6H153H, 190 ± 3 pN ($n$ = 232) for SpecAI88-49E102K-6H149H, and 173 ± 3 pN ($n$ = 204) for SpecAI89-25E48K-24H135H, and an unchanged $\Delta Lc$ of ~53 nm (*Table 2*). These results are consistent with a short, inner-interface coordination clamp that reinforces the native unfolding pathway without introducing alternative topologies.

We further characterized the unfolding kinetics of the computationally designed proteins via dynamic force spectroscopy across a range of pulling speeds, focusing on the SpecAI41 series due to its highest unfolding force. The unfolding forces of the three SpecAI41 exhibited the expected linear dependence on the logarithm of the loading rate (*Figure 4—figure supplement 4*). Fitting the data to the Bell–Evans model yielded a consistent distance to the transition state ($\Delta x \approx 0.19$ nm) for all three constructs, while the spontaneous unfolding rate ($k_{off}$) decreased progressively from SpecAI41 (0.44 $s^{-1}$) to the salt-bridge (0.09 $s^{-1}$) and metal-coordination (0.05 $s^{-1}$) variants. This kinetic profile further confirms that the engineered stabilizations raise the activation barrier for unfolding without altering the unfolding pathway.

Similarly, we assessed the thermal stability of the Stage II variants (*Figure 4—figure supplement 5*). The results demonstrate that the engineered salt bridges and metal-binding sites maintained excellent thermodynamic stability with $T_m$ above 100°C. In chemical stability tests, the Stage II variants still retained considerable helical content under elevated temperature, but their tolerance to GdnHCl decreased slightly compared to the Stage I variants.

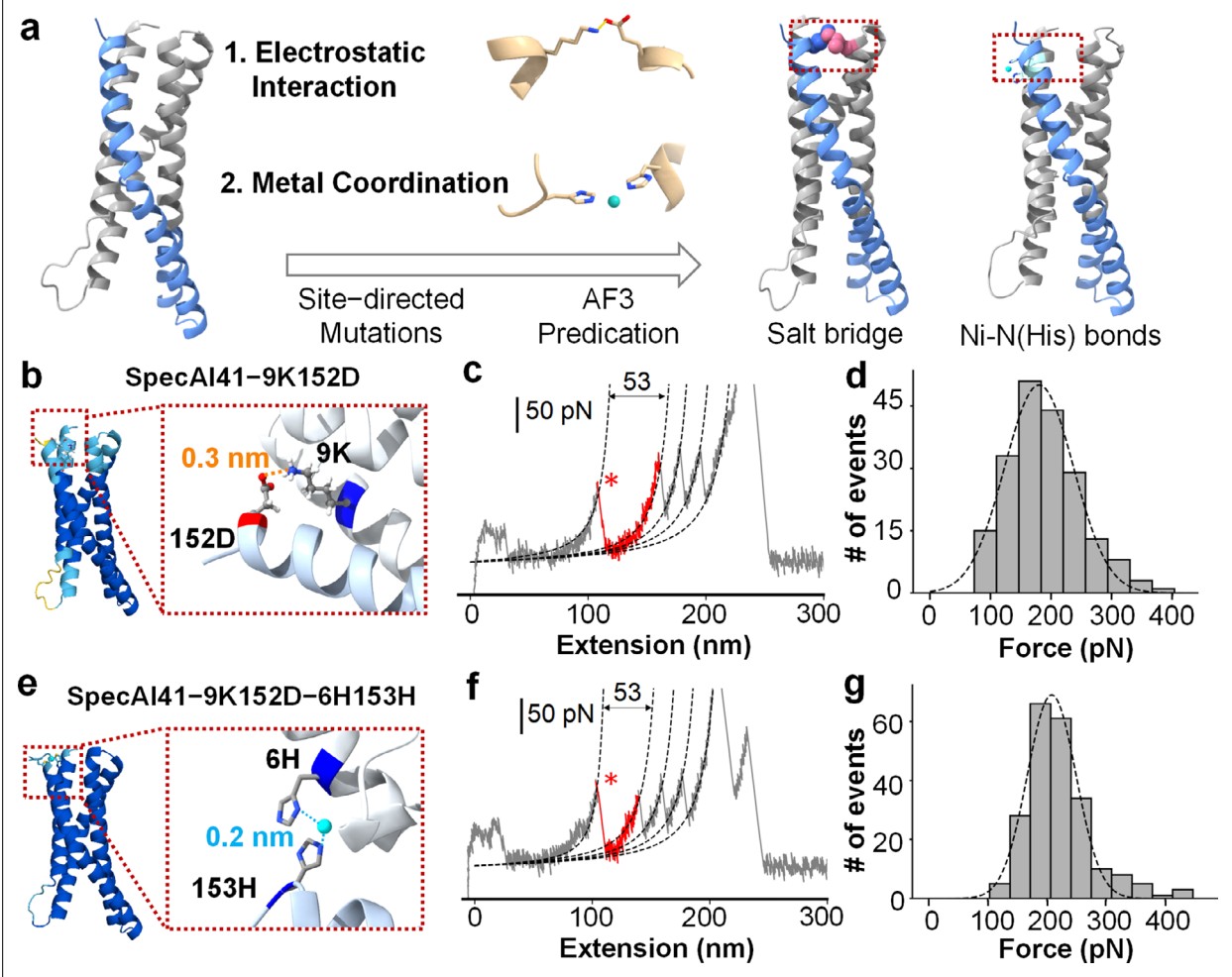

**Figure 4.** Precision stabilization of designed proteins via electrostatic interactions and metal coordination. (**a**) Schematic of Stage II design: introducing inter-helical ion pairs and metal-coordination sites into AI-designed backbones to stabilize specific interfaces. (**b**) AlphaFold3-predicted structure of variant SpecAI41-9K152D, showing an engineered salt bridge (Lys9–Asp152, 3 Å) designed for electrostatic stabilization without perturbing the core. The color of protein is based on the pLDDT value, showing a high-confidence structure prediction (>80). (**c**) A representative force–extension curve for the salt-bridge variant shows an unfolding event with a ΔLc of 53 nm, consistent with the parent scaffold. (**d**) Unfolding force histograms reveal a ~25 pN increase in mechanical stability for salt-bridge variants compared to their Stage I parents, with forces reaching 180 ± 4 pN ($n$ = 197). (**e**) Introduction of a metal-binding site in variant SpecAI41-9K152D-6H153H, with two histidines positioned 2 Å apart, compatible with $Ni^{2+}$ coordination. (**f**) A representative force–extension curve recorded in 200 µM $Ni^{2+}$ shows the unfolding event (ΔLc ≈ 53 nm). (**g**) Unfolding force histograms confirm enhanced mechanical stability, with forces reaching 208 ± 3 pN ($n$ = 222) , a significant gain over the salt-bridge parent.

The online version of this article includes the following source data and figure supplement(s) for figure 4:

**Figure supplement 1.** SDS–PAGE results of SpecAI variants with salt bridge, show their successful expression with expected MW of ~19 $k$Da.

**Figure supplement 1—source data 1.** Original SDS-PAGE gel.

**Figure supplement 1—source data 2.** Original SDS-PAGE gel indicating the relevant bands.

**Figure supplement 2.** SDS–PAGE gel results of SpecAI variants with bis-histidine mutation, show their successful expression with expected MW of ~19 $k$Da.

**Figure supplement 2—source data 1.** Original SDS-PAGE gel.

**Figure supplement 2—source data 2.** Original SDS-PAGE gel indicating the relevant bands.

**Figure supplement 3.** Engineered salt bridges and metal coordination enhance mechanical stability in SpecAI88 and SpecAI89.

**Figure supplement 4.** Dynamic force spectroscopy reveals the kinetic stabilization of the SpecAI41 series.

**Figure supplement 5.** Thermal and chemical stability profiles of Stage II constructs.

**Table 2.** SpecAI stability further enhanced by site-specific functionalization.

| Protein | Force (pN) | ΔLc (nm) | ΔF (pN) | Total ΔF |
|---|---|---|---|---|
| SpecAI88-49E102K | 141 ± 3 | 54 ± 1 | 25 | 74 |
| +6H149H | 190 ± 3 | 53 ± 1 | 49 | |
| SpecAI41-9K152D | 180 ± 4 | 53 ± 2 | 24 | 52 |
| +6H153H | 208 ± 3 | 53 ± 2 | 28 | |
| SpecAI89-25E48K | 153 ± 4 | 53 ± 1 | 32 | 52 |
| +24H135H | 173 ± 3 | 53 ± 1 | 20 | |

Total ΔF means the unfolding force increase over SpecAI.

This hierarchical computational design methodology (*Figure 5*), progressing from architectural stabilization to precision functionalization, enabled the predictable enhancement of protein stability through the additive contribution of distinct chemical interaction networks.

## Discussion

The rational design of proteins capable of withstanding concurrent thermal, chemical, and mechanical stress remains a fundamental challenge. Here, we present a hierarchical strategy that integrates chemically distinct stabilization mechanisms within a single protein architecture. Our approach not only provides a robust blueprint for creating ultrastable proteins but also decouples the quantitative contributions of hydrophobic packing, electrostatic interactions, and metal coordination to mechanical resilience.

Our work builds upon established principles of α-helical protein stability while introducing substantive advances. Whereas most isolated α-helical domains unfold below 20 pN due to sequential hydrogen bond rupture (*Wolny et al., 2014*), our designed variants achieve forces up to 200 pN. This represents an advance beyond canonical α-helical bundles such as spectrin and ankyrin repeats, which

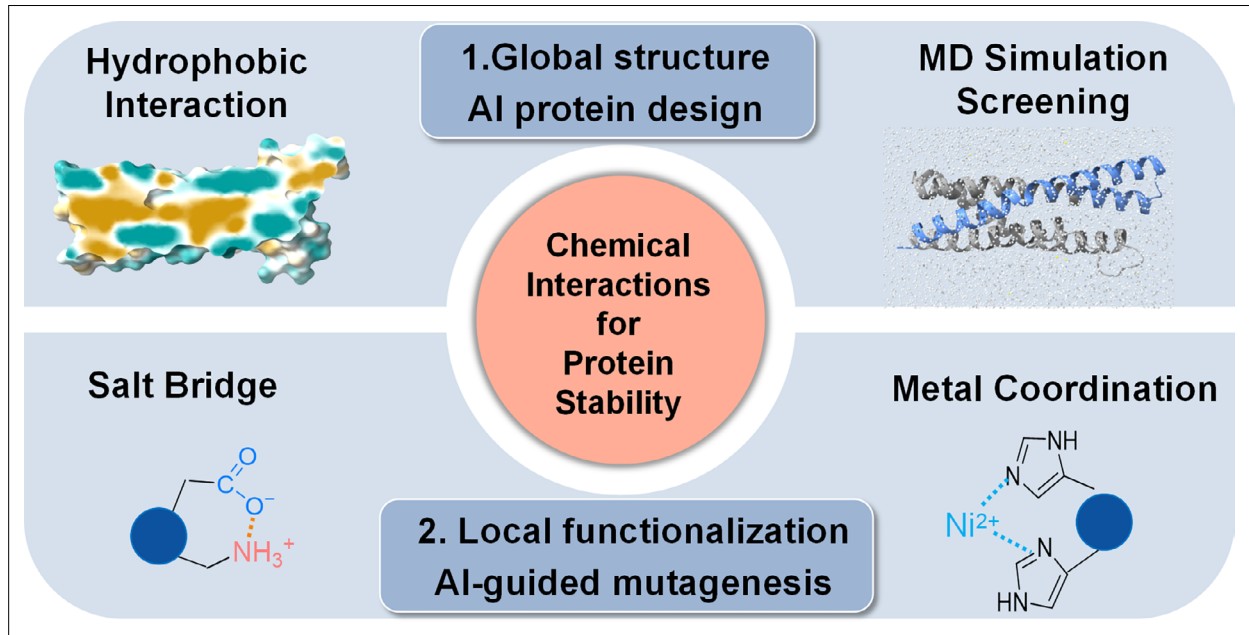

**Figure 5.** Hierarchical computational design of ultrastable proteins through multi-scale stabilization. Schematic summarizing the two-stage design strategy for additive mechanical reinforcement. Stage I establishes a stable architectural framework through molecular dynamics (MD) screening of AI-designed hydrophobically optimized cores. Stage II introduces precision functionalization via inter-helical salt bridges and bi-histidine metal-coordination motifs, guided by AlphaFold3 structural models. The integration of global architectural stability with local chemical cross-linking produces additive mechanical reinforcement, enabling the creation of ultrastable protein domains.

typically unfold near 50 pN (*Serquera et al., 2010*). Previous engineering strategies have demonstrated promising improvements potential (*López-García et al., 2021*), and our contribution lies in the systematic integration of global architectural optimization with local chemical reinforcement within a unified design framework enabled by AI protein design and computational screening (*Thomas and Elcock, 2004*; *Wolny et al., 2017*).

The hierarchical organization of our approach proved effective. We first established a stable structural framework through computational design of an optimized hydrophobic core, providing baseline mechanical stability of ~120 pN. Subsequent precision functionalization with inter-helical salt bridges contributed an additional ~30 pN, while metal-coordination sites provided further gains of ~50 pN depending on geometric parameters. This modular strategy demonstrates that distinct chemical interaction networks can be engineered to operate additively within a single protein structure (*Cao et al., 2011*).

Notably, conventional structural metrics such as pLDDT scores, while predictive of foldability, show limited correlation with mechanical stability. This divergence underscores that dynamic functional properties require physics-based simulation. Although SMD overestimates absolute unfolding forces due to computationally mandated high pulling speeds (*Merkel et al., 1999*) (e.g., ~600 pN in silico vs. ~50 pN for spectrin in experiment), it served as a powerful semi-quantitative screen. Together, these results indicate that SMD simulations provide a useful qualitative reference for comparing relative mechanostability among designs and for guiding experimental prioritization, despite intrinsic differences in force scales and sampling between simulations and experiments.

In conclusion, our hierarchical framework provides a generalizable route to programming protein stability. By modular integration of architectural and chemical stabilization strategies, we can now tailor mechanical, thermal, and chemical resistance in α-helical proteins. This approach will accelerate the development of robust protein-based materials and molecular devices for demanding applications in biotechnology and synthetic biology (*Klein and Nanda, 2025*; *Ling et al., 2018*).

## Materials and methods
### Plasmids construction and proteins engineering

The gene fragments encoding the designed proteins were synthesized (General Biosystems, Anhui) and cloned into the pET-30a vector in the format *Coh-GB1-SpecAI-GB1-His$_6$-NAL* for use in AFM-SMFS experiments (*Shi et al., 2022*). All point mutation constructs introducing electrostatic and metal–ligand interactions were generated by PCR-based site-directed mutagenesis. A construct, *pET30a-SpecAI-His$_6$*, was prepared via seamless cloning for CD measurements.

Both protein constructs were expressed and purified following identical protocols. The plasmids were transformed into *Escherichia coli* BL21(DE3) and plated on Luria-Bertani (LB) agar medium (2.5% LB, 2% agar) containing 100 µg/ml kanamycin. Single colonies were inoculated into 20 ml of LB broth supplemented with kanamycin and incubated overnight at 37°C. These were subsequently expanded into 800 ml of LB medium and grown until the optical density at 600 nm (OD$_{600}$) reached 0.7–0.8. After cooling to room temperature, protein expression was induced by adding 0.5 mM IPTG, followed by incubation at 18°C for 20 hr. Cells were harvested by centrifugation at 4000 rpm for 10 min and stored at −80°C prior to purification.

For purification, cell pellets were resuspended in 30 ml of lysis buffer (50 mM Tris, pH 7.4) supplemented with DNase I, RNase A, and PMSF. Bacterial disruption was carried out using a high-pressure homogenizer (700 bar, 3 min), and the lysate was centrifuged at 13,000 rpm for 15 min to collect the supernatant. This was incubated with Co-NTA affinity resin pre-equilibrated with binding buffer (20 mM Tris, 400 mM NaCl, 2 mM imidazole, pH 7.4) for 30 min at room temperature. After washing with 3–5 column volumes of binding buffer, the target protein was eluted with 15 ml of elution buffer (20 mM Tris, 400 mM NaCl, 250 mM imidazole, pH 7.4).

The *Coh-GB1-SpecAI-GB1-His$_6$-NAL* protein was concentrated via ultrafiltration and buffer-exchanged into a low-salt storage buffer (50 mM Tris, 100 mM NaCl, pH 7.4). The *SpecAI-His$_6$* variant was similarly processed and exchanged into a CD assay buffer (10 mM K$_2$HPO$_4$, 1.8 mM KH$_2$PO$_4$, pH 7.4).

The *GL-His$_6$-GB1-Xmod-Doc*, *GL-His$_6$-ELP20-Cys*, and *His$_6$-Cys-ELP20-NGL* protein used for subsequent single-molecule force spectroscopy experiments was expressed and purified using a similar

procedure. The expression and purification protocols of *Oa*AEP1(C247A) can be found in *Deng et al., 2019*; *Yang et al., 2017*.

## Structural modeling and MD simulations

First, based on spectrin structure, a total of 100 different protein structures were generated using RFdiffusion by appending new structural elements to either the N- or C-terminus (*Watson et al., 2023*). Among these, five structurally reasonable models exhibiting four helical topologies were manually selected for their potential to improve hydrophobic interactions. For each selected structure, 100,000 sequence variants were generated using ProteinMPNN, using a fixed random seed of 37 to ensure reproducibility (*Dauparas et al., 2022*). 319,747 sequences with GRAVY score below –0.3 were selected. The structural compatibility of the generated sequences was then quickly assessed using ESMFold and compared with the corresponding initial structures predicted by Rfdiffusion (*Lin et al., 2023*). Only 357 sequences exhibiting high structural agreement, defined by pLDDT scores above 90 and RMSD values below 2.0 Å, were further modeled using AlphaFold2 (*Jumper et al., 2021*). During AlphaFold2 modeling, six recycles were performed, with template mode enabled. For each sequence, five structural models were generated and ranked according to their pLDDT scores. Ultimately, only the ones with significantly higher scores were retained, and models with pLDDT >90 and RMSD <2 Å were selected for subsequent analysis, resulting in a total of 211 models.

To assess the mechanical stability of designed proteins, SMD simulations were conducted by GROMACS 2023.2 (single-precision build) (*Van Der Spoel et al., 2005*). The systems were constructed using GROMACS built-in tools, with the CHARMM36m force field and the TIP3P water model applied (*Lee et al., 2016*). Rectangular simulation boxes were used with a minimum protein-box distance of 1.2 nm in all directions; for SMD simulations, the $z$ dimension was extended to the theoretical stretching length (initial N-C distance plus 0.36 nm per stretched residue). All simulations were performed assuming a physiological pH of 7.0, with standard protonation states assigned accordingly. The systems were neutralized with sodium and chloride ions to achieve a physiological NaCl concentration of 0.15 M. Simulations were equilibrated under NVT conditions at 310 K using the V-rescale thermostat, followed by a 5-ns NpT production run with C-rescale pressure coupling (1.0 bar), while maintaining temperature control as described above (*Kim et al., 2019*).

Prior to SMD simulations, each system was first subjected to 5000 steps of energy minimization using the steepest descent algorithm, with all protein backbone atoms positionally restrained. Subsequently, during a 1.0-ns NVT equilibration, these restraints were maintained with a harmonic force constant of 1000 kJ mol$^{-1}$ nm$^2$ to allow for the thorough relaxation of the solvent and ions around the protein. After this, 125-ps equilibrium MD runs and 5-ns production MD simulations were run to stabilize the temperature and pressure, ensuring system equilibrium. During SMD simulations, the N-terminus of each protein was restrained using a harmonic potential with a force constant of 1000 kJ mol$^{-1}$ nm$^2$, while the C-terminus was pulled along the $z$-axis at constant velocity using an umbrella restraint with the same force constant ($k = 1000$ kJ mol$^{-1}$ nm$^2$).

SMD simulations were performed at a pulling speed of 1 nm/ns on the 211 predicted protein structures selected from the previous design step (*Gräter et al., 2005*; *Lu et al., 1998*). Each simulation was repeated three times to ensure reproducibility. Since the designed helical proteins could unfold at multiple sites along their structure, the total extension distance was limited to 65% of the theoretical full length, a value that was previously validated through full-length stretching simulations of individual proteins, which confirmed that no significant unfolding events occurred beyond this distance. During SMD simulations, the C-terminus was harmonically restrained while a harmonic pulling potential was applied to the N-terminus in umbrella mode, directed along the $z$-axis. The pulled group was unconstrained in the $xy$-plane, allowing for lateral relaxation during extension. The top three proteins exhibiting the highest average unfolding forces in SMD simulations were selected as candidate constructs. To validate their thermal stability, AMD simulations were conducted on each protein, with temperature gradually increased from 310 to 473 K over a 1-μs trajectory using a 2-fs time step. Structural deviation was quantified by calculating the RMSD between the final structure and the original predicted conformation. These AMD simulations served to validate the selected proteins, ensuring that the mechanically robust candidates also exhibit favorable thermal stability for downstream applications.

## AFM–SMFS experiments

AFM–SMFS protein unfolding experiments were performed on a ForceRobot300 AFM (JPK) at room temperature. Detailed protocols are provided in the literature (*Liu et al., 2024*). In brief, proteins were immobilized on both glass substrate and AFM cantilever via strain-promoted azide-alkyne cycloaddition (SPAAC) and Michael addition chemistry (*Deng et al., 2019*; *Shi et al., 2022*). Elastin-like polypeptide (ELP20, (VPGXG)$_{20}$) was employed as a spacer and single-molecule signature for the experiment (*Ott et al., 2017*). Prior to the measurements, ELP20 with an *Oa*AEP1(C247A) specific recognition site C-terminal NGL (or NAL) or N-terminal GL (*His$_6$-Cys-ELP20-NGL/NAL, GL-His$_6$-ELP20-Cys*) were separately immobilized on the AFM tips and glass substrates through maleimide-thiol reactions. *Oa*AEP1 catalyzes the formation of a covalent bond between a C-terminal -NGL/-NAL tag and an N-terminal -GL motif (*Deng et al., 2019*; *Yang et al., 2017*). Then GB1-Doc (GB1-Xmodule-dockerin) was linked at cantilevers by enzymatic ligation using *Oa*AEP1 to provide X-module-dockerin (Doc) domains (*Stahl et al., 2012*). *Coh-GB1-SpecAl-GB1-His$_6$-NAL* fusion protein was immobilized on a modified glass. The AFM tip was pressed onto the surface with a force of 400 pN to capture the target protein via the specific and high-affinity Coh–Doc interaction, enabling reliable single-molecule attachment.

AFM–SMFS measurements were conducted in buffer composed of 50 mM Tris-HCl and 100 mM NaCl at pH 7.4. For proteins engineered with metal-coordination sites, 200 µM NiCl$_2$ was supplemented to facilitate Ni$^{2+}$ binding. The tip was retracted at a constant pulling speed of 1000 nm/s if not specified. For dynamic force spectroscopy experiments, pulling speeds of 400, 1000, 2000, and 4000 nm/s were used.

Bruker MLCT-D or MLCT-E cantilevers were used, and each cantilever was calibrated in buffer prior to data acquisition using the thermal noise method (equipartition theorem) via the instrument's built-in software. These curves were analyzed by Igor Pro 6.12 (Wavemetrics). Only traces exhibiting at least three consecutive GB1 unfolding peaks followed by a final high force Coh–Doc rupture peak were selected for further analysis. Unfolding forces and contour length increments (ΔLc) were extracted by fitting each individual peak using the WLC model of polymer elasticity. Additionally, the measured unfolding forces were further calibrated using the fingerprint protein GB1 domain as an internal standard, based on its well-established unfolding force (213 pN at 1000 nm/s) (*Cao et al., 2006*). For each construct, unfolding forces from multiple independent traces were pooled and fitted with a Gaussian function. The mean unfolding force and the SEM were extracted from the fits and used for subsequent comparison and analysis.

To investigate the effect of loading rate on protein unfolding behavior, AFM measurements were performed not only at the retraction speed of 1000 nm s$^{-1}$ but also at constant cantilever retraction speeds of 400, 2000, and 4000 nm s$^{-1}$. For each pulling speed, multiple force–extension curves were collected. For each unfolding event, the loading rate $r$ was calculated as the product of the slope of the force peak and the cantilever retraction speed and calibrated using the unfolding force of the GB1 domain as an internal standard (*Cao et al., 2006*). For each speed, the average $r$ and peak unfolding force $F$ were determined, with standard deviations reported as error bars, and plotted in a force–loading rate scatter plot. The zero-force off-rate ($k_{off}$) and the distance to the transition state (Δx) were subsequently extracted by linear fitting based on the Bell–Evans model (*Evans and Ritchie, 1997*):

$$F = \frac{k_B T}{\Delta x} \ln \left( \frac{\Delta x}{k_{off} k_B T} \right) + \frac{k_B T}{\Delta x} \ln (r)$$

where $k_B$ is the Boltzmann constant and $T$ is the temperature.

## Site-directed mutagenesis

To further improve the mechanical stability of helical proteins, salt-bridge mutations were introduced into the top three candidates from Stage I. Two mutation strategies were employed: introducing salt bridges either at the N- or C-terminus of the protein, or near the regions where force peaks were observed in the SMD simulation. Both strategies targeted areas subjected to mechanical stress. Subsequently, we explored metal coordination as an additional stabilization strategy. Double-histidine mutations were introduced at the N- and C-termini of the parent proteins that exhibited high unfolding forces in previous experiments. Structural predictions using AlphaFold3 confirmed that the overall conformations remained stable and that the histidine residues were properly positioned to satisfy the

geometric requirements for $Ni^{2+}$ coordination. These mutated proteins were then subjected to single-molecule force spectroscopy experiments following the same protocol described above.

## MALDI-TOF MS

Measurements were performed on an ultrafleXtreme mass spectrometer (Bruker Daltonics). Protein samples were prepared using a two-layer matrix method at a concentration of approximately 3 mg/ml. Each sample was analyzed in at least three independent replicates to ensure data reliability. Mass spectra were acquired in the $m/z$ range of 18,000–22,000.

## Circular dichroism

CD measurements were performed on a Chirascan (Applied Photophysics) instrument using a 1- mm pathlength cuvette (Hellma). The purified protein was diluted to 0.1–0.15 mg/ml in CD assay buffer. Far-UV CD spectra were recorded from 190 to 260 nm at 20°C to evaluate the secondary structure of SpecAI proteins. To assess thermal stability, temperature-dependent CD measurements were conducted, with spectra collected at 5°C intervals from 20 to 100°C. Thermal unfolding was monitored by tracking the mean residue ellipticity at 195 nm (a characteristic positive peak for α-helices) across a temperature gradient from 20 to 100°C.

For the chemical denaturation study of SpecAI, thermal scanning was conducted in the presence of GdnHCl at concentrations ranging from 1 to 5 M. A set of 4–6 concentrations was selected based on their ability to reliably determine the sample's $Tm$ and produce high-quality fitting curves. Thermal denaturation was performed by heating from 20°C upward in 5°C increments until complete denaturation was achieved. The terminal temperature varied depending on both the specific sample and the concentration of GdnHCl.

At each temperature plateau, full spectra were recorded between 200 and 260 nm. Changes in signal were monitored at approximately 222 nm across all temperatures and GdnHCl concentrations. All measurements were repeated in at least three independent replicates to ensure reproducibility.

## Acknowledgements

The numerical calculations in this work have been done on the computing facilities in the High-Performance Computing Center (HPCC) of Nanjing University.

## Additional information

### Funding
No external funding was received for this work.

### Author contributions
Jun Qiu, Formal analysis, Investigation, Writing – original draft; Guojin Tang, Investigation, Methodology, Writing – original draft; Tianfu Feng, Data curation, Investigation, Writing – original draft; Bin Zheng, Data curation, Formal analysis, Investigation; Yuanhao Liu, Investigation; Peng Zheng, Conceptualization, Supervision, Project administration, Writing – review and editing

### Author ORCIDs
Jun Qiu ⓘ https://orcid.org/0009-0006-5042-9878
Guojin Tang ⓘ https://orcid.org/0009-0001-4899-3692
Tianfu Feng ⓘ https://orcid.org/0009-0001-6261-7298
Bin Zheng ⓘ https://orcid.org/0000-0002-0721-7099
Yuanhao Liu ⓘ https://orcid.org/0009-0009-4892-8290
Peng Zheng ⓘ https://orcid.org/0000-0003-4792-6364

Reviewer #1 (Public review): https://doi.org/10.7554/eLife.109753.3.sa1
Reviewer #2 (Public review): https://doi.org/10.7554/eLife.109753.3.sa2
Reviewer #3 (Public review): https://doi.org/10.7554/eLife.109753.3.sa3

Author response https://doi.org/10.7554/eLife.109753.3.sa4

---

## Additional files

### Supplementary files
MDAR checklist

Source data 1. Candidate proteins identified through ESMFold screening, followed by AlphaFold2 structural prediction and subsequent molecular dynamics simulations.

Supplementary file 1. Protein sequences for AFM-SMFS measurement.

### Data availability
All data were included in the main text or supplementary information.

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
