## [Editor Report · eLife Assessment]

This **important** work describes systematic computational and experimental approaches to turn a moderately stable α-helical bundle into a very stable fold. The authors advance our understanding of α-helix stabilization providing a convenient framework that has general implications for the protein design field. The main claims have **convincing** support through a sound methodology, with strong specific conclusions for designing mechanically, thermally, and chemically stable α-helical bundles.

---

## [Referee Report · Reviewer #1 (Public review)]

Summary:

In the work from Qiu et al. a workflow aimed at obtaining the stabilization of a simple small protein against mechanical and chemical stressors is presented.

Strengths:

The workflow makes use of state-of-the-art AI-driven structure generation and couples it with more classical computational and experimental characterizations in order to measure its efficacy.

The work is well presented and results are thorough and convincing.

The Methods description is quite precise, and some important details were added during review.

Weaknesses:

The pulling velocity is quite high but in accordance with this observation the results were only used for comparative analyses.

Following the review process the authors have shown that the minimum distance between each protein from its periodic images was consistently above 1 nm, yet towards the end of some simulations the value crosses the non-bonded interaction cut-off distance.

Comments on revisions:

The authors did a good job in addressing the reviews.

---

## [Referee Report · Reviewer #2 (Public review)]

Summary:

Qiu, Jun et. al., developed and validated a computational pipeline aimed at stabilizing α-helical bundles into very stable folds. The computational pipeline is a hierarchical computational methodology tasked to generate and filter a pool of candidates, ultimately producing a manageable number of high-confidence candidates for experimental evaluation. The pipeline is split into two stages. In stage I, a large pool of candidate designs is generated by RFdiffusion and ProteinMPNN, filtered down by a series of filters (hydropathy score, foldability assessed by ESMFold and AlphaFold). The final set is chosen by running a series of steered MD simulations. This stage reached unfolding forces above 100pN. In stage II, targeted tweaks are introduced - such as salt bridges and metal ion coordination - to further enhance the stability of the α-helical bundle. The constructs undergo validation through a series of biophysical experiments. Thermal stability is assessed by CD, chemical stability by chemical denaturation, and mechanical stability by AFM.

Strengths:

A hierarchical computational approach that begins with high-throughput generation of candidates, followed by a series of filters based on specific goal-oriented constraints, is a powerful approach for a rapid exploration of the sequence space. This type of approach breaks down the multi-objective optimization into manageable chunks and has been successfully applied for protein design purposes (e.g., the design of protein binders). Here, the authors nicely demonstrate how this design strategy can be applied to successfully redesign a moderately stable α-helical bundle into an ultrastable fold. This approach is highly modular, allowing the filtering methods to be easily swapped based on the specific optimization goals or the desired level of filtering.

Weaknesses:

Assessing the change in stability relative to the WT α-helical bundle is challenging because an additional helix has been introduced, resulting in a comparison between a three-helix bundle and a four-helix bundle. Consequently, the appropriate reference point for comparison is unclear. A more direct and informative approach would have been to redesign the sequence of the original α-helical bundle of the human spectrin repeat R15, allowing for a more straightforward stability comparison.

The three constructs chosen are 60-70% identical to each other, either suggesting over-constrained optimization of the sequence, or a physical constraint inherent to designing ultrastable α-helical bundles. It would be interesting to explore whether choosing a different combination of filters would enable ultrastable α-helical bundles constructs with a more varied sequence content.

While the use of steered MD is an elegant approach to picking the top N most stable designs, its computational cost may become prohibitive as the number of designs increases or as the protein size grows, especially since it requires simulating a water box that can accommodate a fully denatured protein.

Comments on revisions:

The authors have done a good job of addressing the comments.

---

## [Referee Report · Reviewer #3 (Public review)]

Summary:

Qiu et al., present a hierarchical framework that combine AI and molecular dynamic simulation to design α-helical protein with enhanced thermal, chemical and mechanical stability. Strategically chemical modification by incorporating additional α-helix, site-specific salt bridges and metal coordination further enhanced the stability. The experimental validation using single-molecule force spectroscopy and CD melting measurements provide fundamental physical chemical insights into the stabilization of α-helices. Together with the group's prior work on super-stable β strands (https://www.nature.com/articles/s41557-025-01998-3), this research provides a comprehensive toolkit for protein stabilization. This framework has broad implications for designing stable proteins capable of functioning under extreme conditions.

Strengths:

The study represents a complete frame work for stabilizing the fundamental protein elements, α-helices. A key strength of this work is the integration of AI tools with chemical knowledge of protein stability.

The experimental validation in this study is exceptional. The single-molecule AFM analysis provided a high-resolution look at the energy landscape of these designed scaffolds. This approach allows for the direct observation of mechanical unfolding forces (exceeding 200 pN) and the precise contribution of individual chemical modifications to global stability. These measurements offer new, fundamental insights into the physicochemical principles that govern α-helix stabilization.

Weaknesses:

(1) While the initial manuscript lacked a detailed explanation for the stabilizing effect of the additional helix, the revised version now includes a clear structural basis for this improvement. The authors successfully attribute the increased unfolding force threshold to the reinforcement of the hydrophobic core and enhanced cooperative interactions, supported by relevant literature correlations between helix bundle size and stability.

(2) The author analyzed both thermal stability and mechanical stability. It would be helpful for the author to discuss the relationship between these two parameters in the context of their design. Since thermal melting probes equilibrium stability (ΔG), while mechanical stability probes the unfolding energy barriers along pulling coordinate. While the integrative design approach successfully improved both stability types, a deeper exploration of how the specific structural modifications influence the unfolding energy barrier relative to the overall equilibrium stability would further strengthen the mechanistic impact of the work.

(3) While the current study demonstrates a dramatic increase in global stability, the analysis focuses almost exclusively on the unfolding (melting) process. However, thermodynamic stability is a function of both folding (kf) and unfolding (ku) rates. The author have clarified that the observed ultrastability likely originates from a significantly reduced unfolding rates, a hypothesis consistent with the unfolding force. Direct measurements of the kinetics would provide deeper insights.

(4) The authors chose the spectrin repeat R15 as the starting scaffold for their design. R15 is a well-established model known for its "ultra-fast" folding kinetics, with folding rates (kf ~105s), near three orders of magnitude faster than its homologues like R17 (Scott et.al., Journal of molecular biology 344.1 (2004): 195-205). Measuring the folding rates of newly designed proteins would provide additional insights into the design.

Comments on revisions:

I think the author have addressed comments.

---

## [Author Response]

The following is the authors’ response to the original reviews.

**Reviewer #1 (Public review):**
Summary:In the work from Qiu et al., a workflow aimed at obtaining the stabilization of a simple small protein against mechanical and chemical stressors is presented.Strengths:The workflow makes use of state-of-the-art AI-driven structure generation and couples it with more classical computational and experimental characterizations in order to measure its efficacy. The work is well presented, and the results are thorough and convincing.

We are grateful to this reviewer for his/her thoughtful assessment and supportive feedback. In response, we have addressed each comment and incorporated the necessary revisions into the manuscript.

Weaknesses:I will comment mostly on the MD results due to my expertise.The Methods description is quite precise, but is missing some important details:(1) Version of GROMACS used.

We used GROMACS version 2023.2 (single-precision). All subsequent MD simulation procedures mentioned below have been consolidated and described in detail in the Supporting Information (SI).

(2) The barostat used.

Pressure coupling was applied using the C-rescale barostat (τ_p_ = 5.0 ps, ref_p_ = 1.0 bar).

(3) pH at which the system is simulated.

No explicit pH was defined during system construction. Proteins were modeled using standard protonation states as assigned by GROMACS preprocessing tools, corresponding to physiological, near-neutral pH (~ 7.0).

(4) The pulling is quite fast (but maybe it is not a problem)

The relatively high pulling velocity (1 nm/ns) was selected to enable efficient screening across a large number of designed proteins (211 candidates), while maintaining reasonable computational cost/time. Given the intrinsic orders-of-magnitude difference between simulation and experimental pulling rates, SMD results were used as a comparative screening tool, rather than for direct quantitative comparison with AFM data.

(5) What was the value for the harmonic restraint potential? 1000 is mentioned for the pulling potential, but it is not clear if the same value is used for the restraint, too, during pulling.

All positional restraints used in the simulations, including those applied during equilibration as well as the harmonic restraint on the N-terminus and the pulling umbrella restraint during SMD, employed the same force constant (k = 1000 kJ·mol^–1^·nm^2^). We have clarified this point in the revised Methods section.

(6) The box dimensions.

Rectangular simulation boxes were used throughout. For equilibrium MD simulations, the box dimensions in each direction were set based on the maximum extent of the protein along that axis, with a minimum distance of 1.2 nm between the protein surface and the box boundary on all sides. For SMD simulations, the same box dimensions were applied in the x and y directions. Along the pulling (z) direction, the box length was extended to accommodate the theoretical stretching length, defined as the initial N–C terminal distance plus 0.36 nm per stretched residue, while maintaining a 1.2 nm buffer at both ends (2.4 nm total). These details have now been clarified in the revised Supporting Information.

From this last point, a possible criticism arises: Do the unfolded proteins really still stay far enough away from themselves to not influence the result?

We analyzed the minimum atomic distance between each protein and its periodic images to assess potential artifacts from periodic boundary conditions. For all simulation stages used in screening and statistical analysis, the minimum protein–image separation remained above 1.0 nm for the majority of the simulation time, exceeding the nonbonded interaction cutoff and minimizing cross-boundary interactions. As shown in the Author response image 1for SpecAI89 (left), this separation during SMD simulations is consistently well above the threshold, indicating that the chosen box dimensions are appropriate. In the very late stages of annealing MD, highly unstable proteins may exhibit large conformational fluctuations and transient boundary proximity (right); however, these regimes are associated with large RMSD deviations and are excluded from analysis. Notably, the mechanically relevant unfolding events occur near the center of the simulation box and proceed along the pulling axis in SMD simulations, making boundary effects unlikely to influence the unfolding process or the relative mechanostability ranking.

**Author response image 1. sa4fig1:** Analysis of the minimum atomic distance between the protein and its periodic images under periodic boundary conditions. Left: SpecAI89 during SMD simulations, showing that the minimum protein–image distance remains above 1.0 nm for the majority of the simulation time. Right: WT during AMD simulations, where transient proximity to the periodic boundary is observed at very late stages due to large conformational fluctuations.

Additionally, no time series are shown for the equilibration phases (e.g., RMSD evolution over time), which would empower the reader to judge the equilibration of the system before either steered MD or annealing MD is performed.

To assess equilibration, we analyzed the backbone RMSD evolution during the equilibration phase. Using SpecAI89 as a representative example (Author response image 2), the protein backbone RMSD converges rapidly and reaches a stable plateau within approximately 5 ps. The subsequent 125 ps equilibration period therefore sufficiently demonstrates that the system is well equilibrated prior to both steered MD and annealing MD simulations.

**Author response image 2. sa4fig2:** The backbone RMSD of SpecAI89 over time during simulation.

**Reviewer #1 (Recommendations for the authors):**
(1) In Figure S2, only one copy (or the average of the three copies; it is not clear from the caption) is shown, would be better to show the individual traces for each repeat. Additionally, only the plot for the forces is shown, and not, similarly to the AMD, the RMSD plot. This could be a stylistic choice, but it just reports on how much force was applied and not on how the protein responded to the force. Moreover, horizontal lines at the maximum value reached by the force could be added in order to directly see the difference in force applied, since it is then remarked on.

Figure S2 originally shows a representative single SMD trajectory, as the force–extension peak positions vary between independent simulations and averaging the force traces would obscure the characteristic force peaks. In the revised Supplementary Information, we have now added the force–extension traces from the other two independent SMD repeats for each construct (New Figure S2). In addition, horizontal lines indicating the maximum force reached in each trajectory have been included to facilitate direct comparison of force differences between designs.

(2) In Figure S3 the plots have different y-axis. Maybe it could be valuable to modify it so that in figures b, c, and d the spectrum result is in the background (perhaps in gray) so that the y-axis is not changed to retain the information included in this plot, but one could still compare directly to the spectrum result. With a 0 to 1 nm y-axis part of the spectrin run will be hidden, but in any case, plot a can be used to see the full behavior. Similarly to S2, the repeats (if any) could be shown.

We have revised Figure S3 as suggested. The y-axis is now unified to 0–1.2 nm across all panels. For panels b–d, the natural spectrin trajectory is displayed in light gray in the background for direct comparison. Additionally, three independent MD replicates are now presented for each construct to demonstrate reproducibility.

Finally, minor remarks that could nevertheless improve the paper:

(3) In Figure S7, a bimodal distribution model for the number of events could be used to fit the data better.

Following this advice, we explored the bimodal Gaussian distribution model for fitting the force-event data in Figure S7. Indeed, our analysis showed that a bimodal fit could fit Figures S7 panel f better (as shown in Author response image 3). The two peaks were centered at F_1_ = 190 ± 4 pN and F_2_ = 380 ± 6 pN. Interestingly, the force of the first major peak obtained is the same as the previously fitted value. The second one is double force value which we guess maybe is a bi-molecule stretched for unknown reason. Considering the very few numbers of the second peak and the same force value (190 pN), we decide not to change the unfolding force value in the manuscript.

**Author response image 3. sa4fig3:** The bimodal fit for unfolding force of SpecAI88-49E102K-6H149H show the same 190 pN unfolding for the first peak as previous fit.

(4) The colors in the video are not very intuitive, as the spectrin is shown initially in light blue, but becomes grey in the variants, where light blue is reserved for the additional helix. A counter of elapsed time and/or force/temperature applied could help the readers orient. Maybe it could be useful to produce a video with spectrin and the three variants all shown together?

The videos have been revised to improve clarity and consistency accordingly. In all cases, the original protein scaffold is now shown in gray, while the additional helix in the designed variants is highlighted in blue. Real-time annotations have been added to aid interpretation: the instantaneous temperature is displayed during AMD simulations, and time is shown during SMD simulations. In addition, for ease of comparison, the AMD and SMD results of all four proteins are each compiled into a single combined video, allowing their behaviors to be viewed side by side.

**Reviewer #2 (Public review):**
Qiu, Jun et. al., developed and validated a computational pipeline aimed at stabilizing α-helical bundles into very stable folds. The computational pipeline is a hierarchical computational methodology tasked to generate and filter a pool of candidates, ultimately producing a manageable number of high-confidence candidates for experimental evaluation. The pipeline is split into two stages. In stage I, a large pool of candidate designs is generated by RFdiffusion and ProteinMPNN, filtered down by a series of filters (hydropathy score, foldability assessed by ESMFold and AlphaFold). The final set is chosen by running a series of steered MD simulations. This stage reached unfolding forces above 100pN. In stage II, targeted tweaks are introduced - such as salt bridges and metal ion coordination - to further enhance the stability of the α-helical bundle. The constructs undergo validation through a series of biophysical experiments. Thermal stability is assessed by CD, chemical stability by chemical denaturation, and mechanical stability by AFM.Strengths:A hierarchical computational approach that begins with high-throughput generation of candidates, followed by a series of filters based on specific goal-oriented constraints, is a powerful approach for a rapid exploration of the sequence space. This type of approach breaks down the multi-objective optimization into manageable chunks and has been successfully applied for protein design purposes (e.g., the design of protein binders). Here, the authors nicely demonstrate how this design strategy can be applied to successfully redesign a moderately stable α-helical bundle into an ultrastable fold. This approach is highly modular, allowing the filtering methods to be easily swapped based on the specific optimization goals or the desired level of filtering.

We are thankful for the reviewer’s evaluation and remarks. All comments have been incorporated into our revisions.

Weaknesses:Assessing the change in stability relative to the WT α-helical bundle is challenging because an additional helix has been introduced, resulting in a comparison between a three-helix bundle and a four-helix bundle. Consequently, the appropriate reference point for comparison is unclear. A more direct and informative approach would have been to redesign the original α-helical bundle of the human spectrin repeat R15, allowing for a more straightforward stability comparison.

In our case, a substantial fraction of the hydrophobic region is relatively shallow and partially solvent-exposed in the wild-type R15 α-helical bundle. So, the added fourth helix provides a new hydrophobic packing interface, increasing core burial, packing density, and strengthening the internal load-bearing network. Consistent with this design rationale, rSASA analysis shows that the designed proteins exhibit a higher degree of hydrophobic core burial compared to the wild-type R15. Specifically, the fraction of residues with rSASA < 0.2 exceeds 30% in the designs, compared to 23% in the natural spectrin repeat.

While the authors have shown experimentally that stage II constructs have increased the mechanical stability by AFM, they did not show that these same constructs have increased the thermal and chemical stabilities. Since the effects of salt bridges on stability are highly context dependent (orientation, local environment, exposed vs buried, etc.), it is difficult to assess the magnitude of the effect that this change had on other types of stabilities.

We agree that the effects of salt bridges are highly context-dependent and that different dimensions of stability do not always correlate. Following your suggestion, we evaluated the thermal and chemical stabilities of the Stage II constructs. The experimental results (now added as Figure S9) show that Stage II designs successfully maintain the high thermal stability and resistance to chemical denaturation to different extend. The thermal stability is still as high as the Stage I but the resistance to chemical denaturation is slightly reduced. We have added this result in the manuscript accordingly.

The three constructs chosen are 60-70% identical to each other, either suggesting overconstrained optimization of the sequence or a physical constraint inherent to designing ultrastable α-helical bundles. It would be interesting to explore these possible design principles further.

Yes, the observed sequence convergence likely arises from a combination of intrinsic physical constraints of the protein architecture and the applied design and screening criteria. In particular, the tightly packed hydrophobic core imposes strong constraints on side-chain size, packing complementarity, and the alignment of heptad-like motifs reminiscent of coiled-coil organization, which collectively reduce the accessible sequence space. In addition, the strong selection pressure imposed by foldability and stability filters further promotes convergence toward similar solutions. And we agree with the reviewer that this represents an important direction for future work.

While the use of steered MD is an elegant approach to picking the top N most stable designs, its computational cost may become prohibitive as the number of designs increases or as the protein size grows, especially since it requires simulating a water box that can accommodate a fully denatured protein

Yes, steered MD can become computationally expensive, particularly as the number of designs increases or as protein size grows. Considering the vast pool created by AI, SMD in this work was applied to a relatively small, high-confidence subset of candidates after multiple rounds of rapid prescreening, keeping the overall computational cost manageable. In future applications, this step could be further accelerated by integrating machine-learning–based predictors to improve scalability.

**Reviewer #2 (Recommendations for the authors):**
I am not convinced that the difference in rSASA between the designs and the natural spectrin repeat is meaningful. It would be helpful to report confidence intervals for the rSASA values of the designs to clarify whether any differences are statistically robust. Even if such differences prove statistically significant, it is not clear that they are large enough to be practically meaningful.

In our analysis, rSASA values were calculated from equilibrated MD conformations and were consistently higher for all designed proteins that passed the simulation-based screening compared to the wild-type spectrin repeat. However, we believe that rSASA was used only as a supportive structural descriptor to indicate a trend toward a more compact and better-buried hydrophobic core, rather than as a standalone or decisive metric of stability.

Protein stability is indeed influenced by multiple factors, including hydrogen bonding, salt bridges, metal coordination, and topology-dependent load-bearing interactions, none of which are captured by rSASA alone. Therefore, we agree with the reviewer that differences in rSASA alone should not be overinterpreted as a quantitative measure of protein stability. For this reason, rSASA was not used as a ranking criterion or a predictor of stability, but only as complementary evidence consistent with the overall design rationale and with the experimentally observed stability enhancements.

The claim "The strong agreement between computational rankings and experimental measurements validates this approach for prioritizing designs based on relative mechanostability, offering a practical pipeline to bridge the gap between in silico design and experimental validation." should be substantiated by a citation or a figure. Since the authors have the experimental AFM data and steered MD data, I suggest adding a Spearman correlation plot of the two.

Following this comment, we examined the Spearman rank correlation between SMD-derived unfolding forces and experimentally measured AFM forces (Author response image 4). The resulting correlation was modest (*ρ* = 0.4, *p* = 0.6), which is not unexpected given (i) the large difference in force and timescales between high-speed SMD simulations and single-molecule AFM experiments, and (ii) the limited number of designs and simulation repeats available.

Nevertheless, qualitatively, the difference between the first point from wt-spectrin and the other three specAI is clear. Considering the large computational cost, we only performed three times simulation one each design to balance the accuracy and the cost/time. To avoid overinterpretation, we therefore did not include the correlation analysis in the main text and revised the manuscript to soften claims of strong agreement, emphasizing instead the qualitative and comparative role of SMD in the design pipeline.

**Author response image 4. sa4fig4:** Spearman correlation between SMD and AFM unfolding forces for natural spectrin and SpecAI designs. SMD force (x-axis) versus experimental AFM force (y-axis); each point represents one protein.

**Reviewer #3 (Public review):**
Summary:Qiu et al. present a hierarchical framework that combines AI and molecular dynamics simulation to design an α-helical protein with enhanced thermal, chemical, and mechanical stability. Strategically, chemical modification by incorporating additional α-helix, site-specific salt bridges, and metal coordination further enhanced the stability. The experimental validation using single-molecule force spectroscopy and CD melting measurements provides fundamental physical chemical insights into the stabilization of α-helices. Together with the group's prior work on super-stable β strands (https://www.nature.com/articles/s41557-025-01998-3), this research provides a comprehensive toolkit for protein stabilization. This framework has broad implications for designing stable proteins capable of functioning under extreme conditions.Strengths:The study represents a complete framework for stabilizing the fundamental protein elements, α-helices. A key strength of this work is the integration of AI tools with chemical knowledge of protein stability.The experimental validation in this study is exceptional. The single-molecule AFM analysis provided a high-resolution look at the energy landscape of these designed scaffolds. This approach allows for the direct observation of mechanical unfolding forces (exceeding 200 pN) and the precise contribution of individual chemical modifications to global stability. These measurements offer new, fundamental insights into the physicochemical principles that govern α-helix stabilization.

We appreciate the assessment of our manuscript from this reviewer. We have answered all the comments as follows and modified the manuscript accordingly.

Weaknesses:(1) The authors report that appending an additional helix increases the overcall stability of the α-helical protein. Could the author provide a more detailed structural explanation for this? Why does the mechanical stability increase as the number of helixes increase? Is there a reported correlation between the number of helices (or the extent of the hydrophobic core) and the stability?

In multi-helix bundle proteins, tight interhelical packing leads to the formation of a dense hydrophobic core, which substantially enhances overall structural stability. The introduction of an additional helix does not merely increase helix count, but expands the buried hydrophobic interface, improving packing density and cooperative side-chain interactions in the core. This, in turn, strengthens the internal load-bearing network that resists force-induced unfolding.

From a mechanical perspective, adding a helix also increases topological interlocking among secondary-structure elements, which raises the energetic barrier for unfolding and shifts the unfolding pathway toward more cooperative rupture events, thereby increasing the unfolding force threshold. Consistent with this design principle, pioneering studies have reported a positive correlation between the number of helices (or the extent of the hydrophobic core) in helix bundles and their stability (Lim et al., Structure, 2008, 16:449; Minin et al., J. Am. Chem. Soc., 2017, 139, 16168; Bergues-Pupo et al., Phys. Chem. Chem. Phys., 2018, 20, 29105). Inspired by these works, our AI-protein design study uses the appended helix to reinforce the hydrophobic core rather than simply increasing secondary-structure content.

(2) The author analyzed both thermal stability and mechanical stability. It would be helpful for the author to discuss the relationship between these two parameters in the context of their design. Since thermal melting probes equilibrium stability (ΔG), while mechanical stability probes the unfolding energy barriers along the pulling coordinate.

Thermal and chemical stabilities report on the equilibrium free energy (ΔG), while mechanical stability probes the kinetic unfolding barrier (ΔG‡) along a force-dependent pathway. Their inherent difference makes concurrent improvement in all parameters a non-trivial task, which highlights the importance and success of our integrative design approach.

(3) While the current study demonstrates a dramatic increase in global stability, the analysis focuses almost exclusively on the unfolding (melting) process. However, thermodynamic stability is a function of both folding (*k*_f_) and unfolding (*k*_u_) rates. It remains unclear whether the observed ultrastability is primarily driven by a drastic decrease in the unfolding rate (*k*_u_) or if the design also maintains or improves the folding rate (*k*_f_)?

We agree with the reviewer that thermodynamic stability is determined by both the folding rate (*k*_f_) and the unfolding rate (*k*_u_). In the present study, we did not directly measure folding kinetics, and therefore cannot quantitatively deconvolute the respective contributions of *k*_f_ and *k*_u_ to the observed ultrastability. Based on the design strategy and the experimental observations, we propose that the enhanced stability primarily originates from a substantial reduction in the unfolding rate (*k*_u_), corresponding to an increased unfolding energy barrier. The reinforcement of the hydrophobic core, the introduction of stabilizing interactions such as salt bridges and metal coordination, and the additional helix that increases topological and packing constraints all raise the energetic cost of disrupting key interactions in the folded state.

This interpretation is consistent with the high mechanical unfolding forces observed in both AFM experiments and SMD simulations. In contrast, these stabilizing features are not necessarily expected to accelerate folding and may even modestly increase folding complexity. Addressing folding kinetics explicitly would require dedicated kinetic experiments or simulations, which are beyond the scope of the present work but represent an interesting direction for future studies.

(4) The authors chose the spectrin repeat R15 as the starting scaffold for their design. R15 is a well-established model known for its "ultra-fast" folding kinetics, with folding rates (*k*_f_ ~105s), near three orders of magnitude faster than its homologues like R17 (Scott et.al., Journal of molecular biology 344.1 (2004): 195-205). Does the newly designed protein, with its additional fourth helix and site-specific chemical modifications, retain the exceptionally high folding rate of the parent R15?

We did not directly measure the folding kinetics of the newly designed proteins, and therefore cannot determine whether they retain the exceptionally fast folding rate reported for the parent spectrin repeat R15.

**Reviewer #3 (Recommendations for the authors):**
(1) Please clarify the used Gaussian function to fit the unfolding force distribution (Figure 3-4). In Figure S8, the Bell-Evans model is used to analyze unfolding force. The authors should explain the choice of fitting methods and ensure consistency.

The Gaussian fitting used in Figures 3–4 is intended as a descriptive statistical analysis to summarize the unfolding force distributions and to facilitate direct comparison between different designs. This approach provides a robust estimate of the most probable unfolding force and the distribution width, without invoking a specific physical unfolding model, and is commonly used in single-molecule force spectroscopy for comparative purposes.

In contrast, the Bell-Evans model applied in Figure S8 is a kinetic framework that explicitly accounts for force-loading-rate dependence and is used to extract mechanistic insights into the unfolding process. Therefore, the two fitting approaches serve complementary roles: Gaussian fitting for quantitative comparison and ranking of mechanostability, and Bell-Evans analysis for mechanistic interpretation. We have clarified this distinction and the rationale for using both methods in the revised Supplementary Information to ensure consistency and transparency.

(2) The authors utilized steered MD simulation to analyze the mechanical properties via ForceGen (Ni et al., 2024, Sci. Adv. 10, eadl4000). However, the significant discrepancy between the predicted unfolding force (~600 pN) and the experimental value (~50 pN for spectrin, line 376) requires further justification (line 376). Please clarify how the accuracy of these predictions can be established. Specifically, do the MD simulations successfully capture the relative ranking or trends in stability across the different designed variants?

We agree with the reviewer that there is a substantial discrepancy between the absolute unfolding forces predicted by SMD simulations (~ 600 pN) and those measured experimentally by AFM (~ 50 pN for spectrin). This difference primarily arises from the orders-of-magnitude mismatch in loading rates between simulations and experiments. In our SMD simulations, the pulling velocity (~10^9^ nm/s) is several orders of magnitude higher than that used in AFM experiments (~10^3^ nm/s), which is to systematically elevate the apparent unfolding force. In addition to loading-rate effects, limitations in force-field accuracy, finite system size, and restricted conformational sampling further contribute to deviations in absolute force values. As a result, the unfolding forces obtained from SMD are not intended to provide quantitative agreement with experimental measurements or absolute mechanical stability.

Instead, SMD is employed here as a comparative screening tool to assess relative mechanostability across different designed variants under identical simulation conditions. Despite the limited number of repeats imposed by computational cost, the simulations consistently distinguish candidates with markedly different mechanical responses. Importantly, the variants identified by SMD as more mechanically stable were subsequently confirmed experimentally to exhibit enhanced mechanostability relative to the wild-type spectrin repeat. Therefore, while SMD does not yield quantitatively accurate unfolding forces, it successfully captures relative stability trends and provides a practical and effective means for prioritizing designs prior to experimental validation.